# Phonon-enhanced nonlinearities in hexagonal boron nitride

Jared S. Ginsberg[1,11] ✉, M. Mehdi Jadidi[1,11], Jin Zhang [2,11] ✉, Cecilia Y. Chen[3,11], Nicolas Tancogne-Dejean [2], Sang Hoon Chae[4,5,6], Gauri N. Patwardhan[1,7], Lede Xian [2], Kenji Watanabe [8], Takashi Taniguchi [9], James Hone [4], Angel Rubio [2,10] ✉ & Alexander L. Gaeta [1,3] ✉

Polar crystals can be driven into collective oscillations by optical fields tuned to precise resonance frequencies. As the amplitude of the excited phonon modes increases, novel processes scaling non-linearly with the applied fields begin to contribute to the dynamics of the atomic system. Here we show two such optical nonlinearities that are induced and enhanced by the strong phonon resonance in the van der Waals crystal hexagonal boron nitride (hBN). We predict and observe large sub-picosecond duration signals due to four-wave mixing (FWM) during resonant excitation. The resulting FWM signal allows for time-resolved observation of the crystal motion. In addition, we observe enhancements of third-harmonic generation with resonant pumping at the hBN transverse optical phonon. Phonon-induced nonlinear enhancements are also predicted to yield large increases in high-harmonic efficiencies beyond the third.

Parametric optical processes in solids can provide a window into the optical susceptibility, band-structure, and underlying symmetries of crystals, each of which can dramatically affect the nonlinear frequency-conversion process[1–3]. Symmetries, more so than any other factor, dictate the allowed higher-order processes in a given nonlinear system[4]. These properties become frequency independent far from any resonances, as is the case in the visible and near-infrared regime where many high-order harmonic generation measurements take place[5]. However, in the mid-infrared regime, polar crystals support lattice collective oscillations that can be resonantly driven by an optical field. At frequencies near these phonon resonances the linear optical response of the crystal is significantly modified, manifesting for example as a peak in the real permittivity[6,7]. These ionic modes can

alter the symmetry properties of the crystal, leading to transient nonlinear optical effects such as those observed in $SrTiO_3$, which can be driven into a metastable non-centrosymmetric state following prolonged exposure to a phonon-resonant pump[8]. Under increased resonant excitation using femtosecond laser pulses, the amplitude of the ionic motion can become nonlinear with the incident field strength. For bulk materials such as $LiNbO_3$ and GaAs, phonon-induced enhancements of optical nonlinearities[9–12] occur in this regime.

A strong phonon resonance in the mid-IR is present in the van der Waals crystal hexagonal boron nitride (hBN), with a transverse optical (TO) phonon mode at 7.3 μm free-space wavelength (170 meV)[13]. The relatively light constituent atoms of hBN make this one of the most energetic TO phonons, accessible by ultrafast table-top lasers. hBN has

[1]Department of Applied Physics and Applied Mathematics, Columbia University, New York, New York, NY 10027, USA. [2]Max Planck Institute for Structure and Dynamics of Matter and Center for Free-Electron Laser Science, Hamburg 22761, Germany. [3]Department of Electrical Engineering, Columbia University, New York, New York, NY 10027, USA. [4]Department of Mechanical Engineering, Columbia University, New York, New York, NY 10027, USA. [5]School of Electrical and Electronic Engineering, Nanyang Technological University, Singapore 639798, Singapore. [6]School of Materials Science and Engineering, Nanyang Technological University, Singapore 639798, Singapore. [7]School of Applied and Engineering Physics, Cornell University, Ithaca, NY 14853, USA. [8]Research Center for Functional Materials, National Institute for Materials Science, 1-1 Namiki, Tsukuba 305-0044, Japan. [9]International Center for Materials Nanoarchitectonics, National Institute for Materials Science, 1-1 Namiki, Tsukuba 305-0044, Japan. [10]Center for Computational Quantum Physics, Simons Foundation Flatiron Institute, New York, NY 10010, USA. [11]These authors contributed equally: Jared S. Ginsberg, M. Mehdi Jadidi, Jin Zhang, Cecilia Y. Chen. ✉e-mail: jsg2208@columbia.edu; jin.zhang@mpsd.mpg.de; angel.rubio@mpsd.mpg.de; a.gaeta@columbia.edu

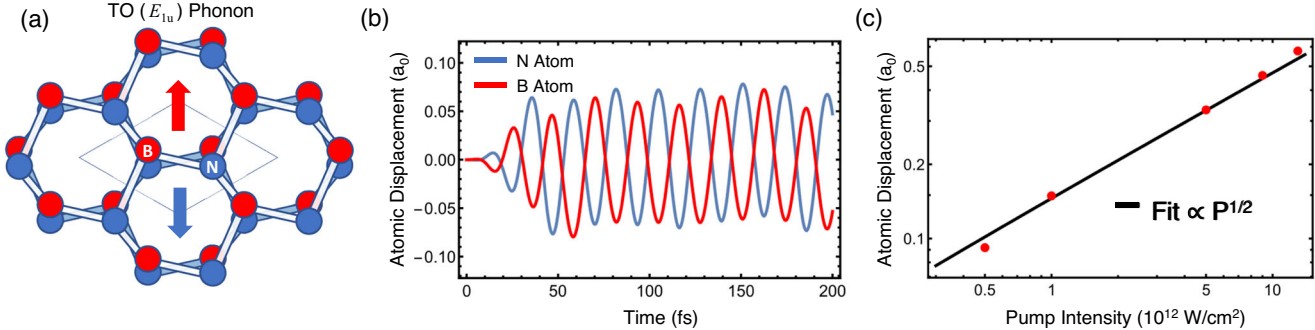

**Fig. 1 | Atomic motion and atomic displacement associated with resonant driving of the TO ($E_{1u}$) phonon mode. a** Honeycomb lattice arrangement of hexagonal boron nitride. Arrows illustrate the motion of atoms under resonant optical excitation. The two species move oppositely from each other in plane and across all layers for the IR-active TO ($E_{1u}$) mode. **b** Simulated atomic displacements, in units of Bohr radii $a_O$, of boron and nitrogen ions in TO ($E_{1u}$)-excited hBN. A 25-fs FWHM, $1 \times 10^{12}$ W/cm$^2$ pulse excites the lattice dynamics. The TDDFT simulations do not include any damping terms through which to estimate the relaxation time. **c** Peak amplitude of atomic displacements as a function of pump intensity, fit to $I^{1/2}$ with a small linear-in-intensity correction. Displacements nearing 10% of the equilibrium lattice constant are achievable before the onset of damage.

an energetically favorable AA' stacked lattice in equilibrium, with alternating boron and nitrogen atoms sitting one on top of the other. An illustration of the resonantly driven, in-plane displacement of atoms for the TO ($E_{1u}$) mode of hBN[14] is presented in Fig. 1a. At the point where the photon and phonon dispersion curves meet, an anti-crossing emerges in the hBN band structure, and the crystal hosts new hybrid modes called phonon-polaritons[15]. These have been the subject of intense study due to their long-range propagation[16].

Using time-resolved measurements, we confirm that when TO phonons of hBN undergo oscillations as indicated by transient four-wave mixing (FWM) signals near the second harmonic generation (SHG) wavelength, which is forbidden with a single beam in a bulk sample at equilibrium[17–19]. The FWM signal is studied as a function of the power and polarization of both the phonon-inducing pump and harmonic-generating probe, from which preferential symmetry axes are identified. Moreover, the natural hyperbolicity of the hBN TO phonon makes it an attractive platform for tight confinement of optical energy, and therefore for enhancing nonlinearities and light-matter interactions within relatively large volumes[6]. We extend the scope of these light-matter interactions to a higher order in mid-IR power by exploiting the strong hyperbolic confinement for even greater electron-phonon coupling. Specifically, in this work we show enhanced emission from the phonon-electron contributions to optical third-harmonic generation (THG) in hBN. We theoretically predict and demonstrate experimentally the nonlinear response of thin hBN crystals associated with this TO phonon mode at 7.3 μm. By sweeping a significant bandwidth of the mid-IR we demonstrate a greatly enhanced on-resonance phononic contribution to THG when hBN is pumped at its TO phonon.

## Results

### Phonon-mediated four-wave mixing

We first characterize theoretically the ionic displacements in bulk hBN under resonant excitation with 25-fs FWHM pulses by performing time dependent density-functional theory (TDDFT) simulated atomic oscillations spanning 200-fs, or roughly 8 times the theoretical pulse duration (see Fig. 1b). For a modest input intensity of $1.5 \times 10^{11}$ W/cm$^2$, we estimate that the phonon amplitude is 1% of the equilibrium lattice constant. While the period of the lattice oscillation is 25-fs, which is consistent with the expected phonon frequency, the relaxation time cannot be theoretically determined due to a lack of dissipative path-ways. The amplitudes of atomic motion are plotted as a function of pump intensity in Fig. 1c. The displacements predicted by TDDFT calculations are fit by $I^{1/2}$ with deviations appearing at large intensities and reach nearly 10% of the equilibrium lattice constant (2.5 Å)[20] at

10 TW/cm$^2$. The time-dependent electronic current is extracted, and from this we generate the theoretical harmonic spectra employed throughout this work (see Methods).

Multilayer hBN has inversion (and 6-fold rotational) symmetry due to the natural 2H stacking of its van der Waals structure[21]. Any contribution at the second harmonic wavelength in few- to many-layer hBN is therefore restricted only to the broken inversion symmetry cases of interfaces and an odd number of layers and is inherently weak[17]. By conducting the ultrafast pump-probe experiments laid out in Fig. 2, we show that excitation of the IR-active TO ($E_{1u}$) phonon allows for the presence of FWM signals at energies of twice the probe photon plus or minus one phonon. Our simulations reveal the emergence of such an ultrafast, transient signal surrounding the second harmonic of an 800 nm probe pulse, as shown in Fig. 3a. The signal on either side of harmonic order 2 highlights the shifting of the signal frequency up and down by the phonon energy in the two variations of the FWM process shown in Fig. 3e. We note that the HHG spectra obtained in the presence of the TO excited phonon display additional signals along with the odd harmonics. This results mostly from the presence of phonon-induced sidebands, which are generated by electron and phonon frequencies (see Supplementary Discussion). The sideband effect also explains the dip at the even harmonic position in our simulations. The energy width between the two split peaks is approximately twice the energy of the TO mode, indicating that the nonlinearity is predominantly third-order.

In our experiments, the measured signal near the second harmonic wavelength of 396 nm is presented in Fig. 3b as a function of the time delay between 792 nm and 7.3 μm pulses. The probe pulse from an amplified Titanium-Sapphire laser is scanned in time by a mechanical delay line relative to the pump pulse from a mid-infrared optical parametric amplifier and difference-frequency generation module. The powers and relative polarizations are set with filters and half-wave plates (HWP), and the two beams are then combined on a beam splitter before being focused onto the sample by a reflective objective. (The experimental setup is shown in Fig. 2b, with further details in Methods.) When the probe pulse precedes the pump pulse, no FWM signal is measured, indicating that the interface SHG and odd layer-number contributions are below the noise floor. The time-resolved signal displays a strong signal at the zero-time delay, when the probe pulse's arrival coincides with the excitation of the hBN phonon-polariton. The transient signal relaxes back to zero with a time constant of 120 fs, which is approximately twice the pump pulse duration. When pumped far off from the phonon resonance, no FWM signal is measured. Fast oscillations on the pump-probe trace provide a direct measurement of the oscillating atomic displacements in time.

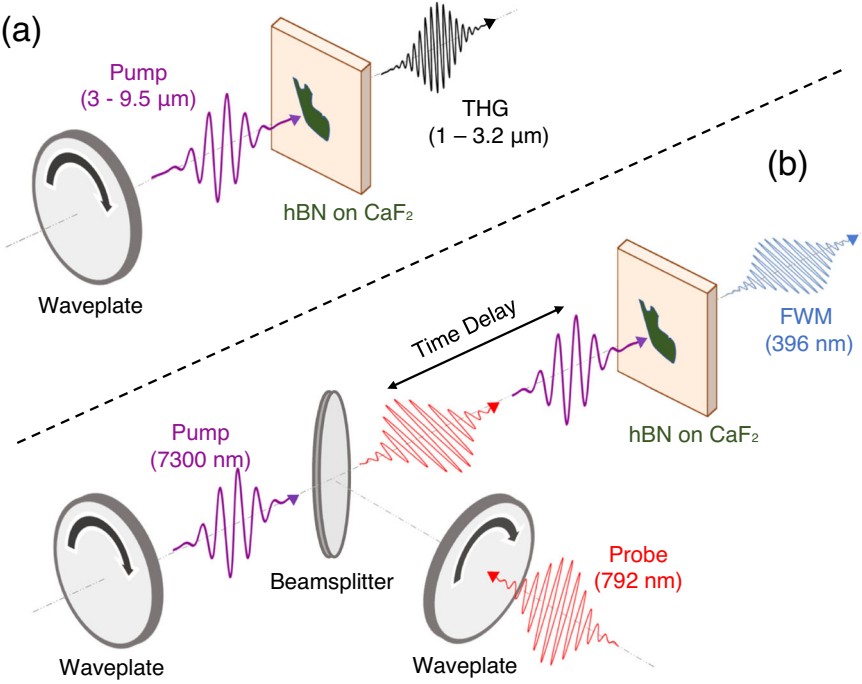

**Fig. 2 | Setup for two experiments demonstrating phonon-enhanced non-linearity in hBN, in transmission geometry. a** Experimental setup for THG experiments. Detection is performed with PbS and MCT detectors, a lock-in amplifier, and boxcar-averaging. **b** Experimental setup for pump-probe FWM experiments. The time-delay is controlled by a mechanical delay stage with sub-1 μm step size. The pump and probe are both focused onto the sample with a reflective objective with 0.5 numerical aperture. Detection is performed with a silicon photomultiplier tube and lock-in amplifier.

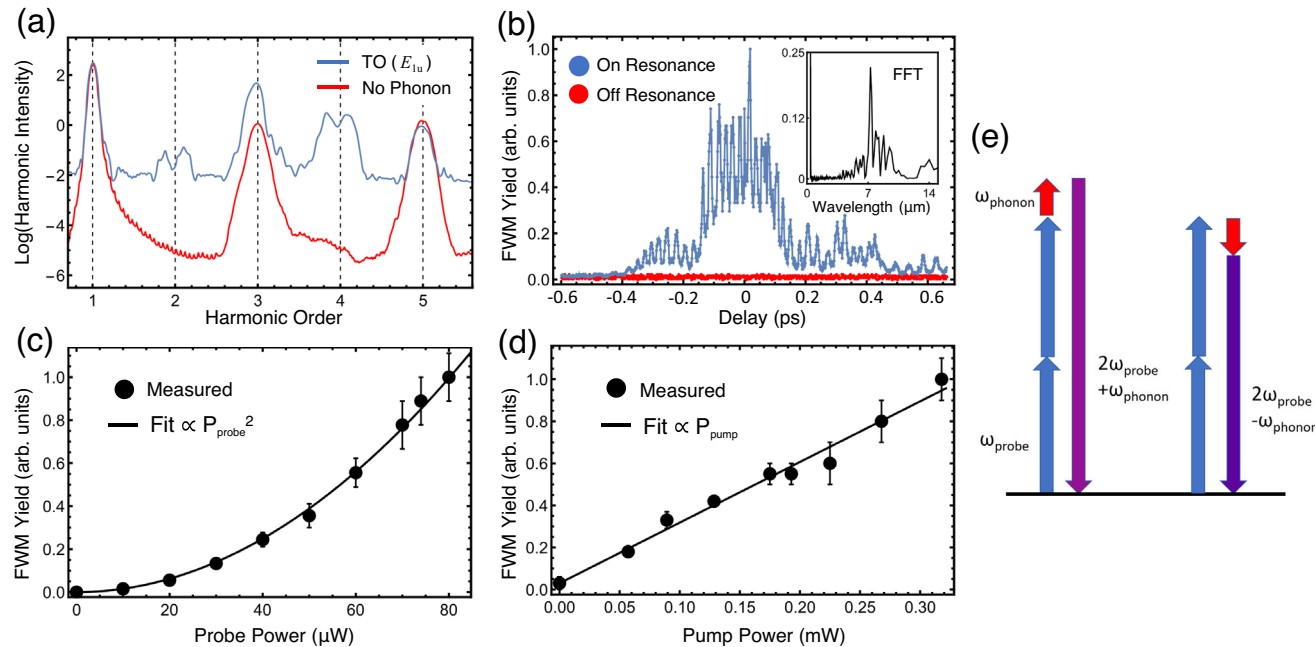

**Fig. 3 | Four-wave mixing between a probe signal and the mid-IR pump at the phonon frequency. a** TDDFT simulations show the emergence of FWM non-linearity during resonant excitation of the TO phonon mode. **b** Time-resolved FWM yield (normalized) of the 792 nm probe pulse. While the pumps are temporally overlapped, an ultrafast third-order nonlinearity is measured. The transient signal vanishes following a 200-fs time constant, or about twice the pulse duration. The appearance of wings in the time-delay scan is a result of a non-perfectly Gaussian pulse, a result of strong atmospheric absorption. Inset: Fourier-Transform of the FWM time-delay. **c** Dependence of measured FWM yield on probe power. **d** Dependence of measured FWM on pump power. The FWM yield increases linearly with the phonon driving intensity. **e** Two versions of the proposed FWM process. The $2\omega_{phonon}$ energy difference in the emissions is consistent with the splitting observed in the theoretical spectra. The error bars in (**c**, **d**) represent the range of the detected signal over the averaging period.

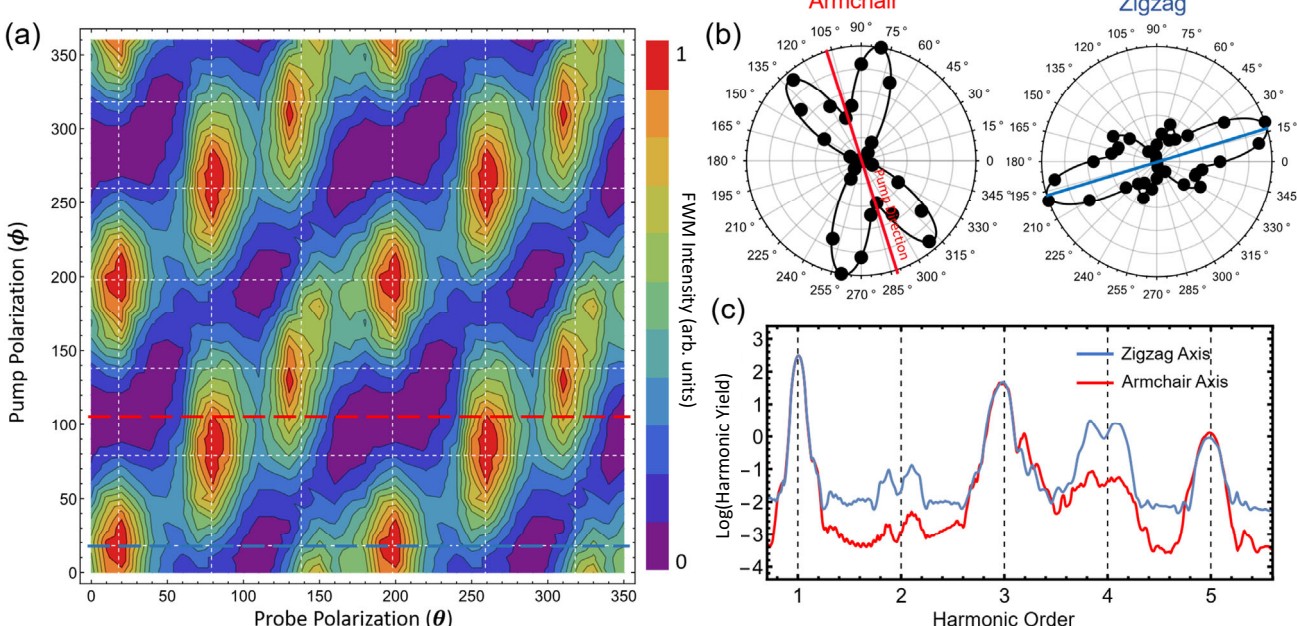

**Fig. 4 | Polarization dependence of the FWM process. a** Pump and probe polarization dependence of total FWM emission (normalized). White dashed lines indicate ZZ axes and are included as a guide for the eye and to emphasize the 60° periodicity. **b** Linecuts along the ZZ and AC axes from (**a**). Solid black lines are fit from Eq. 1, with $\alpha$ and $\beta$ as fit parameters. **c** TDDFT-computed harmonic generation spectra for pump and probe pulses co-polarized along the ZZ (blue) and AC (red) directions. The TO ($E_{1u}$) phonon present along the ZZ axes leads to the greatest nonlinearity.

A pedestal on that same signal is a consequence of the finite response time of each peak being slower than the driving frequency. In Figs. 3c and 3d we show the dependence of the FWM yield on the intensity of the probe and pump, respectively. A quadratic dependence of the FWM intensity on the probe power is observed, while a linear scaling of the signal with respect to the mid-IR intensity is found. Figure 3d reproduces the expected linear dependence on the mid-IR pump pulse based on the pair of $\chi^{(3)}$ interactions depicted in Fig. 3e. We do not observe high-order phonon-resonant processes since the strength of such signals are below the sensitivity of the detection system.

We determine the dependence of the ultrafast FWM on the orientation of the pump and probe polarizations with respect to the crystal high-symmetry axes. Figure 4a gives the total normalized FWM yield for 360° rotation of both pulses (180° rotation is measured and the data is then mirrored). We observe a polarization behavior unique from either the inherent 6-fold $\chi^{(2)}$ or isotropic $\chi^{(3)}$ symmetries of purely electronic hBN nonlinearities[22]. Specifically, the emission appears to closely obey the functional form,

$$FWM(\theta,\phi) = \left[\alpha\cos^2(3\theta) + \beta\sin^2(3\theta)\right]\cos^2(\theta-\phi) \quad (1)$$

where $\theta$ and $\phi$ are the angles of the pump and probe relative to the zigzag (ZZ) axis of the crystal, respectively, and $\alpha$ and $\beta$ determine the relative strengths of the emission along the ZZ and Armchair (AC) axes, respectively. The nonlinear yields peak only along ZZ axes that are being resonantly driven with a phonon-polariton. This is most clearly visible in the linecuts of the probe polarization dependence for pump fields aligned parallel to the AC and ZZ axes, given in Fig. 4b. Even when the pump excitation is aligned with an AC axis, the two adjacent ZZ oriented TO ($E_{1u}$) phonons oscillate with a relatively small amplitude, and we observe phonon-mediated FWM, whereas the ZZ axis at exactly 90° from that excitation shows no emission. From Fig. 4 we determine that the phonon-mediated FWM is at least 3 times greater parallel to ZZ than to the AC directions. This is supported by time-dependent density functional theory simulations in Fig. 4c, which identifies new

nonlinearity along both symmetry axes, though much greater for the TO ($E_{1u}$) phonon than the relative $\pi$ phase LO ($E_{1u}$).

## Phonon-enhanced third-harmonic generation

When driven beyond the previously discussed weak-excitation regime, further enhanced nonlinearities emerge in hBN. We show the integrated and normalized experimental THG amplitudes for a range of pump wavelengths from 3 μm to 9.5 μm in Fig. 5a as blue dots, which are in excellent agreement with the calculations discussed in Fig. 1 and plotted as green dots in Fig. 5a. The third-harmonic exhibits a strong peak for pump wavelengths near the TO phonon resonance at $\lambda = 7.3$ μm, which is far from any electronic or excitonic resonances. We fit the data to a Lorentzian and extract a resonance full-width at half-maximum of 500 nm. THG yields are below the noise level for all $\lambda_{pump} <6$ μm or $>9$ μm, compared to that of the resonant signal which yields at least a 30-fold increase, and thus the phononic enhancement of the THG coefficient at the phonon-polariton wavelength is significantly greater than the purely-electronic component in this regime. In Fig. 5b we plot the measured intensity dependence of the THG signal for $\lambda_{pump} = 7.3$ μm. The fit to a cubic function indicates that the measured nonlinearity is third-order and that the scaling is perturbative, even at high intensities[23]. We note that a similar effect has been observed in the phononic second-harmonic generation of $LiNbO_3$, which also remained in the perturbative regime at higher-than-expected intensities. Ultimately, significant enhancement of the phonon-induced nonlinearity could be further provided through use of subwavelength structures that support confined phonon-polaritons[6,24].

We also performed TDDFT simulations of the wavelength dependence of a higher-order harmonic (HHG) spectra of bulk hBN (see Fig. S1) for two different pump lasers with wavelengths of $\lambda_{pump} = 7.3$ μm (polarized parallel to the TO mode) and $\lambda_{pump} = 6.2$ μm (polarized parallel to the LO mode). Changing the wavelength and polarization of the pump laser can lead to the excitation of different phonon modes and lead to significant modulation of the HHG spectra.

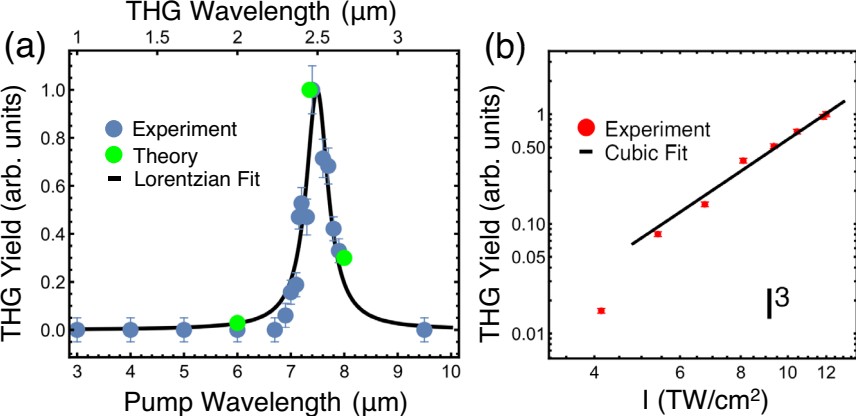

**Fig. 5 | Wavelength and power dependence of the THG process. a** Normalized third-harmonic generation yields of 120-fs pulses as a function of pump wavelengths throughout the mid-IR. THG yields are below the noise level for all wavelengths <6 μm or >9 μm. Within a roughly 1 μm bandwidth of the phonon-polariton resonance, a THG enhancement of 30x is observed. The black line is a Lorentzian fit to the data (blue dots) with full-width at half-maximum of 500 nm. The green dots were obtained by integrating the third-harmonic signal in TDDFT simulations and show excellent agreement with experiments. **b** Normalized intensity dependence of THG pumped on-resonance at 7.3 μm and measured at 2.43 μm. The data is a close fit to $I^3$, indicating that the nonlinear process is scaling perturbatively. Error bars in both figures represent the range of the detected signal over the averaging period.

Excitation of either the TO or LO mode leads to noticeable modifications of the high-harmonic spectra, with the TO ($E_{1u}$) enhancement being one order of magnitude greater than that caused by LO excitation. Furthermore, more intense laser pulses introduce larger phonon amplitudes and lead to larger nonlinearity. As seen from a pump intensity of $2.5 \times 10^{11}$ W/cm², the high-harmonic yields can be increased in a wide energy regime, and the high harmonic generation plateau is enhanced (Fig. S2), which is attributed to the increased atomic movement and enhanced nonlinearity.

## Discussion

We have demonstrated greatly enhanced nonlinearities for optical parametric processes through resonant phonon driving. Furthermore, the appearance of fast oscillations in the pump-probe signal provides the capability for real-time monitoring of atomic motion and evolution driven by ultrafast laser pulses. The maximum achievable FWM efficiency is highly sensitive to the underlying symmetries of the hexagonal lattice, peaking along the ZZ axes where the greatest atomic displacements are known to occur. We extend the light-matter interactions confined by the hyperbolic nature of the hBN phonon dispersion to a strongly nonlinear regime by demonstrating that the large electron-phonon coupling leads to a nearly two order of magnitude enhancement of THG. We note that the phonon resonance present in this work is related to Floquet engineering[25,26]. Floquet engineering involves applying a periodic perturbation to a quantum system, creating a series of states that can be utilized to engineer various properties of the system. The effect of driving the phonon in the harmonic spectra can be interpreted as Floquet phonon engineering, where the harmonic oscillation of the phonon is the external driving frequency in the Floquet theory[26]. Efficient coupling of light to hBN phonon-polaritons at normal incidence places stringent requirements on the allowed optical excitation wavelength. For the free-space wavevector **k** = 0, the required photon wavelength of 7.3 μm is fixed, independent of flake thickness[13]. We only focus on the phonon mode at the Γ point because of energy and momentum conservation. Photon momentum is negligible compared to the size of the Brillouin zone of hBN. The lattice dynamics are driven by an external laser with a wavelength of 7.3 μm polarized parallel to the atomic displacements of the TO ($E_{1u}$) mode. Under those conditions, the simulated real-time evolution of the atomic displacements exhibits clear signatures of only the TO ($E_{1u}$) mode being excited, as shown in Fig. 1b and S1a. Saturation of the THG yield below its perturbative cubic scaling was not observed

and is more likely to occur closer to the onset of sample damage. In a separate theoretical investigation of monolayer hBN, where the TO and LO branches are degenerate at the Γ point, the LO mode was similarly found to yield significant nonlinear effects[27].

## Methods
### Theory
Time-dependent wave functions and electronic currents were computed by propagating the Kohn–Sham equations in real space and real time for TDDFT simulations, as implemented in the Octopus code[28,29]. We employed semi-periodic boundary conditions and the adiabatic LDA[30] functional. All calculations were performed using fully relativistic Hartwigsen, Goedecker, and Hutter (HGH) pseudopotentials[31]. The real-space cell is sampled with a grid spacing of 0.4 bohr and the Brillouin zone is sampled with a 42 × 42 × 21 K-point grid, which yields highly converged results. To model the hBN crystal, the boron nitride bond length is taken here as the experimental value of 1.445 Å. The laser pulses are treated in the dipole approximation using the velocity gauge (implies that we impose the induced vector field to be time-dependent but homogeneous in space), and we use a sine-squared pulse envelope. In all of our calculations, a carrier-envelope phase of $f = 0$ is used[32]. The full harmonic spectra are computed directly from the total electronic current **j**(**r**, t) as

$$HHG(\omega) = \left| FT \left( \frac{\partial}{\partial t} \int d^3 \mathbf{r} \mathbf{j}(\mathbf{r}, t) \right) \right|^2 \qquad (2)$$

where FT denotes the Fourier transform. The calculations are prepared using two independent methods: (i) time-evolution of a distorted atomic configuration (by 1% of the bulk hBN lattice) along the phonon modes; (ii) application of pump laser pulses with the same frequencies and polarizations as the phonon modes. We confirm the two methods are equivalent in the high-harmonic generation simulations.

### Experiments
We conducted the nonlinear experiments on high-quality flakes of hBN, 10–50 nm thick and with sizes on the order of tens of microns. We selected CaF₂ as the preferred substrate for exfoliated flakes based on its high transparency in the visible and mid-infrared ranges and our need for a substrate with low nonlinearity. An amplified Titanium-sapphire laser (Coherent Legend Elite) at 1 kHz repetition rate and 6 mJ pulse energy was used to pump an optical parametric amplifier (OPA,

Light Conversion HE TOPAS Prime). The OPA produces 60-fs duration signal and idler pulses in the near-IR; an additional difference frequency generation (DFG) module seeded by the OPA output provides longer wavelength pulses ($\lambda_{pump}$ = 3–10 μm) with durations of 70 to 120 fs for all mid-infrared measurements. Pulse intensities were kept below the estimated hBN damage threshold of 50 TW/cm². For THG experiments, the pump was focused onto a flake with a 2-cm focal length CaF$_2$ lens. The emitted THG signal was collected in transmission by an identical lens. After rejecting the residual pump beam with a short-pass filter, THG was measured on a PbS detector for $\lambda_{THG}$ below 1.7 μm and on a liquid nitrogen-cooled MCT detector for $\lambda_{THG}$ above 2 μm.

For visible wavelength measurements, the setup was modified to a pump-probe scheme. A 792 nm, 45-fs pulse from the same amplified Ti-Sapphire laser is utilized as the probe. A variable time delay separates the 7.3 μm pump pulse (used to excite the phonon) from the near-IR probe, producing the FWM signal at 396 nm. The intensity of both pulses is kept at or below the TW/cm² range, below the hBN damage threshold. The time delay was controlled by a sub-1-μm step size mechanical delay line. Pump and probe beam polarizations are independently adjusted using zero-order half-wave plates and wire-grid polarizers before being combined on a beam splitter. The collinear pump and probe are focused onto an hBN flake using a reflective objective (NA = 0.5). The choice of reflective optics ensures the same focal plane for the two beams with very different wavelengths. The FWM signal from the 792 nm pulse could be collected by the reflective objective in reflection geometry, or by a CaF$_2$ lens in transmission geometry. The signal was then filtered with a 10-nm bandwidth bandpass filter to reject the residual 792 nm and 7.3 μm light. Detection of the remaining signal was then performed on a fast photomultiplier tube (PMT) and lock-in amplifier.

## Data availability

The data generated during the study is available from the corresponding author upon request.

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

## Acknowledgements

This work is supported as part of Programmable Quantum Materials, an Energy Frontier Research Center funded by the U.S. Department of Energy (DOE), Office of Science, Basic Energy Sciences (BES), under award no. DE-SC0019443. The work of J.Z., LX., N.T.-D., and A.R. was supported by the European Research Council (ERC-2015-AdG694097), the Cluster of Excellence 'CUI: Advanced Imaging of Matter' of the

Deutsche Forschungsgemeinschaft (DFG)—EXC 2056—project ID 390715994, Grupos Consolidados (IT1249-19), partially by the Federal Ministry of Education and Research Grant RouTe-13N14839, the SFB925 "Light induced dynamics and control of correlated quantum systems," The Flatiron Institute is a division of the Simons Foundation. Support by the Max Planck Institute—New York City Center for Non-Equilibrium Quantum Phenomena is acknowledged. J.Z. acknowledges funding from the European Union's Horizon 2020 research and innovation pro-gram under the Marie Sklodowska-Curie grant agreement No. 886291 (PeSD-NeSL). We thank I-Te Lu for the fruitful discussions. K.W. and T.T. acknowledge support from the Elemental Strategy Initiative conducted by the MEXT, Japan (Grant Number JPMXP0112101001) and JSPS KAKENHI (Grant Numbers 19H05790 and JP20H00354). C.Y.C. acknowledges support from the NSF Graduate Research Fellowship Program DGE 16-44869.

## Author contributions

J.S.G., C.Y.C., M.M.J., and G.N.P. performed experiments. J.Z., N.T.-D., and L.X. performed theory and simulations. Samples from K.W. and T.T. were prepared by S.H.C. Research was supervised by A.L.G., A.R., and J.H.

## Competing interests

The authors declare no competing interests.
