## [Peer Review File · Nature Communications]

Phonon-Enhanced Nonlinearities in Hexagonal Boron NitrideREVIEWER COMMENTS

Reviewer #1 (Remarks to the Author):

In this paper, the authors reported the phonon-enhanced nonlinearities in hexagonal boron nitride. They observed enhancements of third-harmonic generation with resonant pumping at the hBN transverse optical phonon. A FWM signal, with two probe photons, one resonant phonon, and one signal photon, was measured. I have certain concerns on the claim of phonon resonance and the written format of the paper. Some of the suggestions and concerns are given below:

- 1, The energy difference between pump photon and probe photon is very large. It is possible to excite some high-order phonon-resonant process? e.g. $2\omega_{\text{probe}} \pm 2\omega_{\text{phonon}}$? $2\omega_{\text{probe}} \pm 3\omega_{\text{phonon}}$?
2. The authors wonder the selection rules for the involved phonon mode. Was only E1u allowed or not?
3. The infrared/Raman spectra of used h-BN sample should be given in Supporting Materials.
4. In my understanding, this phonon resonance is a typical photon-phonon coupling. If sure, maybe only the energy eigenvalue in the center of Brillouin zone ($k=0$) is involved. How about of phonon dispersion of h-BN with large k ? I hope the authors to discuss this case.
5. In Fig. 1c, 3c, 5b, a log-log plot is better to fit the nonlinear intensity. Especially in Fig. 5b, the power density of pump light is very high ($\sim 10 \text{ TW/cm}^2$). In previous report, the high-order (including THG) nonlinearities would deviate from perturbative scaling and exhibit a gradual saturation. I hope the authors check this case. Also, the error bar should be plotted in Fig. 5b.
Ref: Yang, Y., et al. (2019). "High-harmonic generation from an epsilon-near-zero material." Nature Physics 15: 1022-1026.
6. The readers expect to see some direct evidences for phonon resonance. For example, a giant enhancement on infrared absorption or Raman intensity of h-BN with E1u mode. An in-situ experiment under 7300 nm-pumping is better.
7. The last question is only an academic discussion. Is this phonon-resonance present here same to Floquet engineering that far from the equilibrium state?

In summary, I feel that this work is interesting and can be considered for publication in Nature Commun after revisions.

Reviewer #2 (Remarks to the Author):

The manuscript is devoted to experimental probing of cubic nonlinearity enhancement in hexagonal boron nitride (h-BN). By using four-wave mixing and third-harmonic generation the authors detected phonon resonances in the mid-IR spectral range. The time-resolved measurements allow for probing of the crystal motion.

The idea of the manuscript is novel, experimental results are well supported by modelling, the manuscript is well-written. I guess that the manuscript meets Nature Comm criteria for novelty and scientific importance. It might be recommended for publication after revisions reflecting the following critics.

1. The experimental methods are described very accurately. The pulse duration, pulse intensities, wavelengths are presented in detail. However, I haven't found any quantities for nonlinearities. The estimation of absolute values of susceptibilities or nonlinear efficiencies is very appreciated, especially in spectral ranges of phonon resonances.
2. Discussion of TDDFT simulations (Fig.3a) shows the presence of even-order nonlinearities. However, excitation of the TO phonon mode strongly, by several orders of magnitude, enhances the noise level. What is the reason for that? Additionally, the lineshape of even harmonics possesses a dip exactly at the harmonic position. How it can it be understood?
3. Fig.S2 shows high harmonic spectra for different pump laser intensities. The spectra are measured for two, five-times different, values of pump intensities. However, the intensities of harmonic, for say, the 3-rd, the 4-th and the 5-th, appears almost the same. What is the reason for this?
4. Equation 1 mentioned in the caption of Fig.4 is absent.

Reviewer #1 (Remarks to the Author):

In this paper, the authors reported the phonon-enhanced nonlinearities in hexagonal boron nitride. They observed enhancements of third-harmonic generation with resonant pumping at the hBN transverse optical phonon. A FWM signal, with two probe photons, one resonant phonon, and one signal photon, was measured. I have certain concerns on the claim of phonon resonance and the written format of the paper. Some of the suggestions and concerns are given below:

1. The energy difference between pump photon and probe photon is very large. It is possible to excite some high-order phonon-resonant process? e.g., 2ω probe $\pm 2\omega$ phonon? 2ω probe $\pm 3\omega$ phonon?

Reply: From our TDDFT simulations, the HHG spectra obtained in presence of the excited phonons are attributed to two harmonic contents: i) the electron frequencies, which are the integer multiple of the laser field, and ii) the frequency of the phonon (ω_p). This can be explained using the following formula for HHG in solids [Tancogne-Dejean et al., Phys. Rev. Lett. 118, 087403 (2017)]:

$$HHG(\omega) \propto \left| FT \left(\int_{\Omega} dr n(r, t) \nabla v_0(r, t) \right) + N_e E \right|^2,$$

where FT denotes the Fourier transform, Ω is the simulation cell volume containing N_e electrons, $n(r, t)$ is the time-dependent electronic density excited by the electric field E with frequency of ω_0 , and $v_0(r, t)$ the electron-ion potential, which depends on time when phonons are excited. From this formula, the spectra, in presence of a phonon mode oscillating at the frequency ω_p contain a series of peaks at $(2n + 1)\omega_0 + m\omega_p$.

From Figures S1, we do note several smaller peaks, indicating such high-order processes are possible. However, we do not observe high-order phonon-resonant signals (2ω probe $\pm 2\omega$ phonon or 2ω probe $\pm 3\omega$ phonon) since the strength of such processes are weaker than what our system can detect in the experiments. Operating with higher energy pulses to observe these processes resulted in damage to the sample.

To address this comment, we have added the following sentence on page 3: "*We do not observe high-order phonon-resonant processes since the strength of such signals are below the sensitivity of the detection system.*"

In addition, we have added the above detailed discussion in the revised supplementary materials.

2. The authors wonder the selection rules for the involved phonon mode. Was only E1u allowed or not?

Reply: The lattice dynamics are driven by an external laser with a wavelength of 7.3 μm polarized parallel to the atomic displacements of the TO (E_{1u}) mode. Under those conditions, the simulated real-time evolution of the atomic displacements exhibits clear signatures of only the TO (E_{1u}) mode being excited, as shown in Figures 1b and S1a. There are no other modes found in our simulations for this particular laser pulse. In contrast, when we employ a laser with a wavelength of 6.2 μm , the LO (E_{1u}) mode is excited and a different HHG spectrum results, as shown in Figure S1b.

We have added the following sentences in the revised manuscript (Page 6): "*The lattice dynamics are driven by an external laser with a wavelength of 7.3 μm polarized parallel to the atomic displacements of the TO (E_{1u}) mode. Under those conditions, the simulated real-time evolution of the atomic displacements exhibits clear signatures of only the TO (E_{1u}) mode being excited, as shown in Figures 1b and S1a.*"

3. The infrared/Raman spectra of used h-BN sample should be given in Supporting Materials.

Reply: The Raman spectrum of hBN showing its characteristic E_{2g} peak corresponding to a Raman-active in-plane vibration mode has been added to the Supplementary Material as Figure S3.

4. In my understanding, this phonon resonance is a typical photon-phonon coupling. If sure, maybe only the energy eigenvalue in the center of Brillouin zone ($k=0$) is involved. How about of phonon dispersion of h-BN with large k ? I hope the authors to discuss this case.

Reply: The reason why we focus on the phonon mode at the Γ point is mainly due to 1) the energy E conservation between phonons and photons, i.e., $E = \hbar\Omega(q) = \hbar\omega(k)$, where $\Omega(q)$ is the phonon at the wave number q and $\hbar\omega(k)$ is the photon frequency at the photon wave number k ; and 2) the momentum P conservation between phonons and photons, i.e., $\hbar k = \hbar q$. In our case, for the phonon momentum to satisfy both energy and momentum conservation, q should be very close to the Γ point. More specifically, the photon frequency $\omega(k)$ is the same as the phonon frequency $\Omega(q)$, and the photon momentum is $k \approx 1370 \text{ cm}^{-1}$, which is negligible compared to the size of the Brillouin zone where $|G| \approx 2 \times 10^8 \text{ cm}^{-1}$, i.e., $\frac{k}{|G|} \approx 5 \times 10^{-6}$. Therefore, one only needs to focus on the Γ point for phonons.

In the main text, we added the following sentences on Page 6: "*We only focus on the phonon mode at the Γ point because of energy and momentum conservation. Photon momentum is negligible compared to the size of the Brillouin zone of hBN.*" We have also included the more in-depth discussion above in the Supplementary Material.

5. In Fig. 1c, 3c, 5b, a log-log plot is better to fit the nonlinear intensity. Especially in Fig. 5b, the power density of pump light is very high ($\sim 10 \text{ TW/cm}^2$). In previous report, the high-order (including THG) nonlinearities would deviate from perturbative scaling and exhibit a gradual saturation. I hope the authors check this case. Also, the error bar should be plotted in Fig. 5b.

Ref: Yang, Y., et al. (2019). "High-harmonic generation from an epsilon-near-zero material." Nature Physics 15: 1022-1026.

Reply: We have replotted the three figures of interest on log-log axes, attached below as side-by-side comparisons to their original plots on linear axes. We have also reduced the data point marker size to reveal the error bars more clearly (note that some error bars are less visible than others in a log-log plot).

Figure R1: Side-by-side comparison of Figure 1c in the main text, plotted on log-log versus linear axes.

Figure R2: Side-by-side comparison of Figure 3c in the main text, plotted on log-log versus linear axes.

Figure R3: Side-by-side comparison of Figure 5b in the main text, plotted on log-log versus linear axes.

While we agree that log-log plots can reveal the presence of any saturation, we believe that plotting Figure 3c in particular with log-log axes may confuse readers because it is placed next to Figure 3d, which would look extremely similar but is a lower-order function. Figure 3c would be a linear trend line on log-log axes representing the four-wave mixing (FWM) yield with respect to probe power, while Figure 3d would be a linear trend line on linear axes representing the FWM yield with respect to pump power. We have attached the proposed figure below and hope the referee agrees that the juxtaposition of Figures 3c and 3d plotted in this way could mislead readers.

Figure R4: Modified Figure 3 with 3c plotted on log-log axes.

Finally, several examples from the literature on 2D materials do not report harmonic saturation for THG [Ref. 1-2]. We have included two references, in hBN and WSe₂, below. We believe the results we present are in line with existing reports for hBN.

Ref. 1: hBN THG (Figure 3b)

A. Popkova et al., "Optical Third-Harmonic Generation in Hexagonal Boron Nitride Thin Films," ACS Photonics 8, 3, 824–831 (2021).

Ref. 2: WSe₂ THG (Figure 3b)

H. G. Rosa et al., "Characterization of the second- and third-harmonic optical susceptibilities of atomically thin tungsten diselenide," Sci Rep 8, 10035 (2018).

6. The readers expect to see some direct evidences for phonon resonance. For example, a giant enhancement on infrared absorption or Raman intensity of h-BN with E_{1u} mode. An in-situ experiment under 7300 nm-pumping is better.

Reply: The linear reflection spectrum of hBN about the TO (E_{1u}) phonon resonance at 7.3 μm (equivalent to 1370 cm⁻¹) has been added as Supplementary Figure S4. The reflection signal is normalized to the substrate. This spectrum is evidence of a large increase in reflection at the corresponding phonon mode wavelength. This linear

measurement at 7.3 μm complements the nonlinear measurements and TDDFT simulations made at 7.3 μm in the main text.

7. The last question is only an academic discussion. Is this phonon-resonance present here same to Floquet engineering that far from the equilibrium state?

Reply: This is an interesting question. We note that the phonon resonance present in the work is indeed related to Floquet engineering. Both phonon resonance and Floquet engineering involve non-equilibrium states. Phonon modes can be excited at specific frequencies, resulting in phonon amplification and the creation of higher energy phonons. This phenomenon has potential applications in developing phonon-based technologies, including high-frequency electronics and phonon-based quantum computing.

On the other hand, Floquet engineering involves applying a periodic perturbation to a quantum system, creating a series of "Floquet states" that can be utilized to engineer various properties of the system. This technique has potential applications in fields such as quantum computing and quantum simulation.

In short, the effect of driving the phonon in the harmonic spectra can be interpreted as Floquet phonon engineering, where the harmonic oscillation of the phonon is the external driving frequency in the Floquet theory [Ref. 3].

In response to this comment, we added the following sentences on page 5 in the revised manuscript: "*We note that the phonon resonance present in this work is related to Floquet engineering [Ref. 3-4]. Floquet engineering involves applying a periodic perturbation to a quantum system, creating a series of states that can be utilized to engineer various properties of the system. The effect of driving the phonon in the harmonic spectra can be interpreted as Floquet phonon engineering, where the harmonic oscillation of the phonon is the external driving frequency in the Floquet theory [Ref. 3].*"

Ref. 3: H. Hübener, U. D. Giovannini, A. Rubio. Phonon driven Floquet matter. Nano Lett. 18, 2, 1535–1542 (2018).

Ref. 4: A. Castro, U. D. Giovannini, S. A. Sato, H. Hübener, A. Rubio. Floquet engineering the band structure of materials with optimal control theory. Phys. Rev. Res. 4, 033213 (2022).

In summary, I feel that this work is interesting and can be considered for publication in Nature Commun after revisions.

Reply: We thank the referee for her/his constructive comments and recommendations.

Reviewer #2 (Remarks to the Author):

The manuscript is devoted to experimental probing of cubic nonlinearity enhancement in hexagonal boron nitride (h-BN). By using four-wave mixing and third-harmonic generation the authors detected phonon resonances in the mid-IR spectral range. The time-resolved measurements allow for probing of the crystal motion.

The idea of the manuscript is novel, experimental results are well supported by modelling, the manuscript is well-written. I guess that the manuscript meets Nature Comm criteria for novelty and scientific importance. It might be recommended for publication after revisions reflecting the following critics.

Reply: We thank the referee for providing us with insightful and valuable comments and recommendations, especially pointing out: "*The idea of the manuscript is novel, experimental results are well supported by modeling, the manuscript is well-written.*"

1. The experimental methods are described very accurately. The pulse duration, pulse intensities, wavelengths are presented in detail. However, I haven't found any quantities for nonlinearities. The estimation of absolute values of susceptibilities or nonlinear efficiencies is very appreciated, especially in spectral ranges of phonon resonances.

Reply: Given the nanometer-scale thickness of our samples and the micron-scale wavelengths central to our mid-IR phenomena in hBN, a complex technique of spectrally resolved two-beam coupling (SRTBC) is necessary to measure nonlinear susceptibility with sufficient sensitivity. However, based on direct prior experience with SRTBC in thin and two-dimensional materials [Ref. 5-6], we estimate that such an experiment would encompass the scope and novelty of a separate paper. We can, however, extrapolate a lower bound on the value of $\chi^{(3)}$ from our data for Figure 5a.

The peak THG intensity for a pump wavelength of 7.3 μm is 16.6 times greater than the minimum measurable off-resonance THG signal (corresponding to a pump wavelength of 6.9 μm). To our knowledge, measurements of $\chi^{(3)}$ in hBN have not been performed in the mid-IR. Popkova et al. [Ref. 1] reported a third-order susceptibility value of $8.4 \times 10^{-21} \text{ m}^2/\text{V}^2$ for a wavelength of 1080 nm in the near-infrared. Using this value, we can give an order-of-magnitude estimate for the lower bound of $\chi^{(3)}$ on resonance as $10^{-19} \text{ m}^2/\text{V}^2$.

We have included the following sentences below Figure S4 in the Supplementary Material: "*From Figure 5a, the peak THG intensity for a pump wavelength of 7.3 μm is 16.6 times greater than the minimum measurable off-resonance THG signal (corresponding to a pump wavelength of 6.9 μm). To our knowledge, measurements of $\chi^{(3)}$ in hBN have not been performed in the mid-IR. Popkova et al. [Ref. 1] reported a third-order susceptibility value of $8.4 \times 10^{-21} \text{ m}^2/\text{V}^2$ for a wavelength of 1080 nm in the near-infrared. Using this value, we can extrapolate an order-of-magnitude estimate for the lower bound of $\chi^{(3)}$ on resonance as $10^{-19} \text{ m}^2/\text{V}^2$.*"

Ref. 1: A. Popkova et al., "Optical Third-Harmonic Generation in Hexagonal Boron Nitride Thin Films," *ACS Photonics* 8, 3, 824–831 (2021).

Ref. 5: G. Patwardhan et al., "Nonlinear refractive index of solids in mid-infrared," *Opt. Lett.* 46, 1824-1827 (2021).

Ref. 6: G. Patwardhan et al., "Gate-Tunable Kerr Nonlinearity of Graphene in the Mid-Infrared," *2020 Conference on Lasers and Electro-Optics (CLEO), San Jose, CA, USA*, pp. 1-2 (2020).

2. Discussion of TDDFT simulations (Fig.3a) shows the presence of even-order nonlinearities. However, excitation of the TO phonon mode strongly, by several orders of magnitude, enhances the noise level. What is the reason for that? Additionally, the lineshape of even harmonics possesses a dip exactly at the harmonic position. How it can be understood?

Reply: We agree that the HHG spectra obtained in the presence of the TO excited phonon display additional signals along with the odd harmonics. This results mostly from the presence of phonon-induced sidebands, which are generated by two harmonic contents: i) the electron frequencies, which are the integer multiples of the laser field, and ii) the frequency of the phonon (ω_p). This can be explained using the following formula for HHG in solids [Tancogne-Dejean et al., *Phys. Rev. Lett.* 118, 087403 (2017)]:

$$HHG(\omega) \propto \left| FT \left(\int_{\Omega} dr n(r, t) \nabla v_0(r, t) \right) + N_e E \right|^2,$$

where FT denotes the Fourier transform, Ω is the simulation cell volume containing N_e electrons, $n(r, t)$ is the time-dependent electronic density excited by the electric field E with frequency ω_0 , and $v_0(r, t)$ is the electron-ion potential, which depends on time when phonons are excited. From this formula, the spectra, in the presence of a phonon mode oscillating at frequency ω_p contain a series of peaks at $(2n + 1)\omega_0 + m\omega_p$.

The sideband effect also explains the dip at the even harmonic position in our simulations. The energy width between the two split peaks is approximately twice the energy of the TO mode, indicating that the nonlinearity is predominantly third-order. Figure S1a further confirms the importance of the phonon-induced oscillation since a peak is present in the spectrum at 0.17 eV, which corresponds to the energy of the vibrations of the TO (E_{1u}) mode in hBN (where electrons adiabatically follow the lattice). For a more general perspective, we explore the HHG spectrum with a driven longitudinal optical (LO) mode in Figure S1b. This demonstrates that the spectra contain mode-selective information.

We have added the following on page 2 of the main text: "*We note that the HHG spectra obtained in the presence of the TO excited phonon display additional signals along with the odd harmonics. This results mostly from the presence of phonon-induced sidebands, which are generated by electron and phonon frequencies (see discussion under Figure S1). The sideband effect also explains the dip at the even harmonic position in our simulations. The energy width between the two split peaks is approximately twice the energy of the TO mode, indicating that the nonlinearity is predominantly third-order.*"

In addition, we have added the above detailed discussion in the revised supplementary materials.

3. Fig. S2 shows high harmonic spectra for different pump laser intensities. The spectra are measured for two, five-times different, values of pump intensities. However, the intensities of harmonic, for say, the 3-rd, the 4-th and the 5-th, appears almost the same. What is the reason for this?

Reply: We agree the figure is unclear. Here, we replot Figure S2 in linear scale to emphasize the difference between the peaks.

Figure R5: HHG spectra for different pump laser intensities in linear scale. Here, the polarizations of the pump and probe lasers are parallel with the TO (E_{1u}) mode (pump laser with a wavelength of $\lambda = 7300 \text{ nm}$). For the probe laser, we use an in-plane driving electric field with a wavelength of $\lambda = 800 \text{ nm}$ and an intensity of $I = 10^{12} \text{ W/cm}^2$, and a pulse duration of 25-fs full width at half maximum.

We would like to clarify that the powers being modified in the two spectra of Figure S2 are the pump powers, while the harmonic signals are being generated by the probe pulse which is held constant at 10^{12} W/cm^2 . The odd-order harmonics are almost entirely probe power dependent (not pump-mediated). Hence, one should not expect to see large changes in the harmonic yields as the pump power is increased. In the two spectra above with pump intensities of $0.5 \times 10^{11} \text{ W/cm}^2$ and $2.5 \times 10^{11} \text{ W/cm}^2$, the 3rd and 5th harmonic intensities (integrated over an energy window of $E_{\text{harm}} \pm \frac{1}{2} E_{\text{probe}}$) increased by factors of 1.5 and 1.0, respectively. By comparison, the novel sideband effects are driven by the pump power, and we observe a relatively larger 4.3-times enhancement of the signal near the second harmonic frequency.

To make the figure clear, we have updated Figure S2 and related discussions in the Supplementary Material.

4. Equation 1 mentioned in the caption of Fig.4 is absent.

Reply: On page 4, we added the equation as follows:

"Specifically, the emission appears to closely obey the functional form,

$$FWM(\theta, \phi) = [\alpha \cos^2(3\theta) + \beta \sin^2(3\theta)] \cos^2(\theta - \phi) \quad , \quad (1)$$

where θ and ϕ are the angles of the pump and probe relative to the zigzag (ZZ) axis of the crystal, respectively, and α and β determine the relative strengths of the emission along the ZZ and Armchair (AC) axes, respectively. Sideband yields peak only along ZZ axes that are being resonantly driven with a phonon-polariton. This is most clearly visible in the linecuts of the probe polarization dependence for pump fields aligned parallel to the AC and ZZ axes, given in Figure 4b."

REVIEWERS' COMMENTS

Reviewer #1 (Remarks to the Author):

The revised manuscript has addressed my comments.

Reviewer #3 (Remarks to the Author):

The manuscript entitled "Phonon-Enhanced Nonlinearities in Hexagonal Boron Nitride" by Ginsberg et al. is a theoretical and experimental investigation on the phononic nonlinearity of hBN. The results are very interesting, sound and original, and to my knowledge, they represent the first experimental evidence of phononic nonlinearity in hBN in the mid-IR. It is remarkable that there are no saturation effects due to nonlinear absorption or higher-order nonlinearities (χ^5 and χ^7 , etc). Indeed, the I^3 cubic fit is preserved up to TW/cm^2 peak intensities. I have only a couple of suggestions:

1. is there a way to provide an estimate of the absorbance near the polaritonic resonance - I see that in fig. S4 there is a reflectivity peak, and this is due to the increase of the real part of the χ^1 , but what about the imaginary part of χ^1 ?
2. I would like to suggest including the following reference: <https://doi.org/10.1021/acsnano.1c03775> in which a theoretical investigation of the polaritonic nonlinearity of monolayer hBN is reported.

In conclusion, the results are very interesting and manuscript is well written and clearly reports the experimental setup and the numerical methods (TDDFT). Therefore I would like to recommend publication of the manuscript.

Reviewer #1 (Remarks to the Author):

The revised manuscript has addressed my comments.

Reply: Thank you for your feedback.

Reviewer #3 (Remarks to the Author):

The manuscript entitled "Phonon-Enhanced Nonlinearities in Hexagonal Boron Nitride" by Ginsberg et al. is a theoretical and experimental investigation on the phononic nonlinearity of hBN. The results are very interesting, sound and original, and to my knowledge, they represent the first experimental evidence of phononic nonlinearity in hBN in the mid-IR. It is remarkable that there are no saturation effects due to nonlinear absorption or higher-order nonlinearities (χ_5 and χ_7 , etc.) Indeed, the I^3 cubic fit is preserved up to TW/cm^2 peak intensities. I have only a couple of suggestions:

1. Is there a way to provide an estimate of the absorbance near the polaritonic resonance - I see that in fig. S4 there is a reflectivity peak, and this is due to the increase of the real part of the χ_1 , but what about the imaginary part of χ_1 ?

Reply: The absorbance of hBN near the polaritonic resonance can be obtained with a measurement of the transmission spectrum. We attempted to measure the transmission of the sample, but at this time, we do not have access to a system that can produce precise and accurate measurements from thin, small-area exfoliated flakes of hBN on a transparent substrate.

We believe a more precise and accurate measurement of the reflection spectrum about the polaritonic resonance in a sample of thin exfoliated hBN flakes on a reflective substrate can be found in a more recent paper of ours that is currently under review [Ref. 1], in Figure 1b. Therefore, we have removed the original reflection spectrum, Figure S4 in the supplementary material, and added the following reference:

Ref. 1: Chen, C. Y. et al. Unzipping hBN with ultrashort mid-infrared pulses. Preprint at <https://doi.org/10.48550/arXiv.2205.12310> (2022).

2. I would like to suggest including the following reference: <https://doi.org/10.1021/acsnano.1c03775> in which a theoretical investigation of the polaritonic nonlinearity of monolayer hBN is reported.

Reply: We have included the above paper in the references and added a sentence in the manuscript at the end of the Discussion section: *“In a separate theoretical investigation of monolayer hBN, where the TO and LO branches are degenerate at the Γ point, the LO mode was similarly found to yield significant nonlinear effects.”*